# Diesel Exhaust Particulates Enhances Susceptibility of LPS-Induced Acute Lung Injury through Upregulation of the IL-17 Cytokine-Derived TGF-β_1_/Collagen I Expression and Activation of NLRP3 Inflammasome Signaling in Mice

**DOI:** 10.3390/biom11010067

**Published:** 2021-01-06

**Authors:** Dong Im Kim, Mi-Kyung Song, Kyuhong Lee

**Affiliations:** 1National Center for Efficacy Evaluation of Respiratory Disease Products, Korea Institute of Toxicology, 30 Baehak1-gil, Jeongeup 56212, Korea; dongim.kim@kitox.re.kr (D.I.K.); mikyung.song@kitox.re.kr (M.-K.S.); 2Department of Human and Environmental Toxicology, University of Science & Technology, Daejeon 34113, Korea

**Keywords:** diesel exhaust particulate, acute lung injury, lung fibrosis, NLRP3, IL-17, TGF-β_1_

## Abstract

Diesel exhaust particulates (DEP) adversely affect the respiratory system and exacerbate lung diseases, resulting in high mortality rates. However, its pathogenesis is complicated, and the mechanisms involved are incompletely understood. We investigated the effects of DEP pre-exposure on lipopolysaccharide (LPS)-induced acute lung injury (ALI) and identified the roles of interleukin (IL)-17 in mice. Mice were divided into vehicle control, DEP, LPS, and DEP pre-exposed and LPS-instilled groups. Pre-exposure to DEP enhanced the number of total cells, neutrophils, and lymphocytes in the BAL fluid of LPS-instilled mice. Pre-exposure to DEP synergistically exacerbated pulmonary acute lung inflammation and granulomatous inflammation/pulmonary fibrosis, concomitant with the enhanced expression of inflammatory cytokines in the BAL fluid and of collagen I and TGF-β_1_ in the lungs of LPS-instilled mice. The number of TGF-β_1_-positive cells in the DEP pre-exposed and LPS-instilled group was higher than that in the LPS group. The expression of NLR family pyrin domain containing 3 (NLRP3) inflammasome components was markedly increased in the DEP pre-exposed and LPS-instilled group. IL-17 levels in the BAL fluid and IL-17-positive cells in the lungs were significantly increased by pre-exposure to DEP in the LPS-induced group compared to that in the DEP or LPS group. These results suggest that DEP predominantly contributes to fibrotic lung disease in LPS-related acute lung injury by upregulating IL-17 cytokine-mediated collagen I and TGF-β_1_ and, at least in part, by activating LPS-induced NLRP3 inflammasome signaling. The study should be useful in devising better strategies for prevention and management of ALI.

## 1. Introduction

Acute lung injury (ALI) and acute respiratory distress syndrome (ARDS) are devastating disorders of the lung that are characterized by lung inflammation, pulmonary edema, low compliance, and capillary leakage due to increased pulmonary vascular permeability. These disorders affect individuals of all ages [1], and the risk of developing ALI depends on patient characteristics. ALI is easily triggered by events, such as pneumonia, gastric aspiration, inhalation of smoke and toxic gases, reperfusion, and severe sepsis [2]. Recently, numerous clinical studies have shown a positive correlation of exposure to particulate matter (PM) with the number of outpatient visits, emergency visits, and hospitalizations for acute upper or lower respiratory infections, indicating an increase in the susceptibility to respiratory infections [3]. PM_2.5_, a subset of PM with a diameter less than 2.5 μm, has a large surface area and can adsorb a variety of toxic and harmful substances. Because of its small size, it can be inhaled with breath and get deposited in the terminal bronchioles and alveoli. Moreover, PM_2.5_ enters the circulatory system through the gas–blood barrier and causes systemic adverse effects [4]. PM_2.5_ such as diesel exhaust particulates (DEP) is considered to be among the most harmful environmental risk factors. Epidemiological studies have shown a consistent association between elevated levels of PM in ambient air and increased respiratory mortality and morbidity. Lin H. et al. has reported that short-term exposures to PM might be important triggers of ARDS in China [5]. Furthermore, Rhee J. et al. has demonstrated that long-term exposures to ambient PM_2.5_ and ozone increased ARDS risk for older adults in the United States [6]. To date, it has reported that exposures to air pollutant are associated with increased risk of ARDS [4,5,6,7,8]. In addition, in an in vivo toxicological study, it was found that exposure to PM_2.5_ including DEP increases susceptibility to lung infection by inducing dysfunction of immune cells [9,10,11]. Some experimental studies have revealed that DEP enhance LPS-induced ALI through the increased expression of proinflammatory cytokines, chemokines, and toll-like receptors [12] as well as through the activation of intercellular adhesion molecule-1 and nuclear factor-κB (NF-κB) p65/p50 [13]. Furthermore, Yanagisawa, R. et al. have demonstrated that DEP synergistically enhance ALI showing upregulation of S100 calcium-binding protein A9, lipocalin 2, and small inducible cytokine B family member 10 though complementary DNA microarray analysis in LPS-induced mice [14]. Despite of many studies regarding the association between air pollution and LPS-induced ALI/ARDS, the mechanisms underlying these observations are not yet completely understood.

IL-17 is produced by Th17 lymphocytes, many innate-like lymphocytes, including γδ T cells, invariant natural killer cells (iNKT), and type 3 innate lymphoid cells (ILC3) [15]. High expression of IL-17 has been linked to inflammatory diseases of the mucosal surface, such as asthma, cystic fibrosis, and chronic obstructive pulmonary disease (COPD) in the airway, as well as to inflammatory bowel disease. Some clinical studies have reported that the levels of IL-17 and other Th17 cytokines are increased in sputum and airways of patients with COPD and might play a role in orchestrating neutrophilic inflammation in the lungs [16,17]. In addition, in vivo studies have revealed that IL-17A exacerbates the severity of lung infection as evidenced by the increase in serum IL-17A levels upon infection with Mycobacterium pulmonis and reduction in disease severity by neutralization of IL-17A in Balb/c mice [18].

The NLR family pyrin domain containing 3 (NLRP3) is activated by a variety of signals, including pathogen-associated molecular patterns (PAMPs), danger (or damage)- associated molecular patterns (DAMPs), and bacterial toxins [19,20,21]. Activation of NLRP3 leads to the assembly of inflammasome, a cytosolic multiprotein complex for the activation of caspase-1, which is required for proteolytic maturation and release of the pro-inflammatory cytokines, IL-1β and IL-18. Although NLRP3 inflammasome activation is critical for driving acute lung inflammation and aids in the clearance of viral and bacterial infections, persistent activation of NLRP3 by irritants leads to the production of IL-1β. Its contribution to the increased risk of secondary bacterial infections after influenza or other viral infections results in the progression of several chronic pulmonary diseases, including pulmonary fibrosis, COPD, ALI/ARDS, and asthma [22]. An association between NLRP3 inflammasome-derived IL-1 and IL-17 with pulmonary inflammation and fibrosis has been reported in the lung inflammation model of mice [23,24,25]. We hypothesized that PM_2.5_ can affect pathophysiological changes in infection-induced ALI by contributing to IL-17 expression and NLRP3 inflammasome activation. 

Airway administration of lipopolysaccharide (LPS), a major pro-inflammatory component of gram-negative bacteria, is a common method for studying pulmonary inflammation and ALI in small animal models [26,27,28,29]. Many toxicological studies have shown that exposure to LPS results in the recruitment of neutrophils and increases the expression of pro-inflammatory molecules in animal models. We recently reported that intratracheal instillation with diesel exhaust particulates (DEP) induces neutrophilic lung inflammation in mice [30,31]. In this study, we investigated whether DEP enhance susceptibility to LPS-induced ALI and examined the role of IL-17 and NLRP3 inflammasomes in the mechanisms underlying such an effect.

## 2. Materials and Methods 

### 2.1. Animals

Female Balb/c mice (Orient Bio, Seongnam, Korea) weighing 15.83 ± 0.56 g were housed in a temperature-controlled room (22 °C ± 3 °C) under a 12/12-h light/dark cycle with free access to standard laboratory chow and tap water. The mice were used for experiments after 8 days of acclimation, during which time they showed no adverse clinical signs and exhibited normal weight gain. The experiments were performed in accordance with protocols approved by the Institutional Animal Care and Use Committee of the Korea Institute of Toxicology (no. 1901-0006).

### 2.2. Study Protocol

The mice were randomly divided into four weight-matched experimental groups (n = 5 each) using the Pristima v.7.3 preclinical software program (Xybion Medical Systems Corporation, Morris Plains, NJ, USA) and treated intratracheally as follows. Mice in the vehicle control (VC) group received 50 μL distilled water (DW) as the DEP control and instilled 50 μL normal saline as LPS control. The DEP group received 100 μg DEP (SRM 2975; National Institute of Standards and Technology, Gaithersburg, MD, USA) dispersed in 50 μL DW and instilled 50 μL normal saline. The LPS group received 50 μL DW and instilled 20 μg LPS (Sigma-Aldrich, St. Louis, MO, USA) dissolved in 50 μL of normal saline to induce ALI. DEP pre-exposure and LPS-instilled groups were pretreated with 100 μg DEP in 20 μg LPS-induced mice. At 48 h after LPS instillation, mice were euthanized with an overdose of isoflurane and continuously exposed until 1 min after breathing stopped. For analysis, bronchoalveolar lavage (BAL) cells, BAL fluid, and lung tissues were collected from the sacrificed animals.

### 2.3. DEP and LPS Instillation

Prior to tracheal injection with vehicle, DEP, and LPS, isoflurane was delivered into the induction chamber using small animal portable anesthesia systems (L-PAS-02, LMSKOREA, Inc., Seongnam, Korea) equipped with an isoflurane vaporizer. The mice were then exposed to 2.5% isoflurane delivered through O_2_ (2 L/min) within the induction chamber until a sleep-like state was reached. Mice receiving isoflurane anesthesia were removed from the induction chamber and instillation was performed immediately on board. Mice were intratracheally instilled with 100 μg DEP on days 1, 4, and 7. LPS (20 μg) was intratracheally instilled on day 7 (Figure 1). In the case of DEP pre-exposed and LPS-instilled groups, LPS was instilled 30 min after the last DEP instillation. After instillation, the mice showed movement and complete recovery, and were transferred to their cage.

### 2.4. Measurement of Body and Organ Weights

The body weight of mice was measured on days 1, 4, 7, and 9. On day 9, mice were sacrificed, and weights of the lungs, spleen, and thymus were recorded.

### 2.5. Preparation of BAL Fluid

At 48 h after LPS instillation, mice were anesthetized with isoflurane and exsanguinated. The left lung was ligated and the right lung was gently lavaged three times via the tracheal tube with a total volume of 0.7 mL phosphate-buffered saline (PBS; Thermo Fisher Scientific, Waltham, MA, USA). The total number of cells in the collected BAL fluid was counted using a NucleoCounter (NC-250; ChemoMetec, Gydevang, Denmark). For differential cell counts, BAL cell smears were prepared using Cytospin (Thermo Fisher Scientific) and were stained with Diff-Quik solution (Dade Diagnostics, Aguada, Puerto Rico). A total of 200 cells per slide were counted. The BAL fluid was immediately centrifuged at 2000 rpm for 5 min, and the collected supernatant was stored at −70 °C until the measurement of cytokine levels by enzyme-linked immunosorbent assay (ELISA).

### 2.6. Measurement of Cytokine Levels

The levels of IL-1β, IL-6, tumor necrosis factor (TNF)-α, and IL-17 in the BAL fluid were quantified by ELISA using commercial kits (Thermo Fisher Scientific), according to the manufacturer’s protocol. The sensitivity of IL-1β, IL-6, TNF-α, and IL-17 assays was 1.2, 6.5, 3.7, and 1.6 pg/mL, respectively.

### 2.7. Histology and Immunohistochemistry

At 48 h after LPS instillation, mice were sacrificed for histologic assessment. The lung tissue was removed and fixed in 10% (*v*/*v*) neutral-buffered formalin, dehydrated, embedded in paraffin, and cut into 4-μm sections that were deparaffinized with xylene and subjected to hematoxylin and eosin (H&E; Sigma-Aldrich) and Masson trichrome (MT) staining. The stained sections were analyzed under a light microscope (Axio Imager M1; Carl Zeiss, Oberkochen, Germany). The degree of inflammation was scored on a scale of 0 to 4 as previously described [32]. Pulmonary fibrosis was evaluated by MT staining by determinig the Ashcroft score [33]. For immunohistochemistry of TGF-β_1_ and IL-17A, the deparaffinized 4-μm sections were incubated sequentially with reagents in accordance with the instructions for the Ready-To-Use Vectastain Universal Quick kit (Vector Laboratories, Burlingame, CA, USA). Briefly, the slides were incubated in Endo/Blocker (Biomeda Corp, Foster City, CA, USA) for 15 min and in proteinase K (Dako, Glostrup, Denmark) for 15 min at 37 °C. They were then incubated in normal horse serum for 30 min at room temperature and probed with anti-TGF-β_1_ (Abcam, Cambridge, MA, USA) and anti-IL-17A (Novus Biologicals, Littleton, CO, USA) antibodies for 2 h at room temperature. Thereafter, the slides were incubated with prediluted, biotinylated panspecific IgG for 30 min. The slides were subsequently incubated in streptavidin/peroxidase complex reagent for 15 min and then in a 3-amino-9-ethylcarbazole substrate kit for 5 min. Controls consisted of sections of the lung tissue from mice incubated without the primary antibody. After immunostaining, the slides were photomicrographed (Axio Imager M1, Carl Zeiss). The degree of immunoreactivity was scored on a scale of 0 to 4, as previously described [34].

### 2.8. Preparation of Protein Extract and Western Blot Analysis

The lung tissue was homogenized in RIPA buffer (Thermo Fisher Scientific) according to the manufacturer’s protocols, and protein concentrations were determined using Bradford reagent (Bio-Rad, Hercules, CA, USA). Proteins were separated by SDS-PAGE at 120 V for 90 min and then transferred to a polyvinylidene difluoride membrane (Bio-Rad) at 250 mA for 90 min by wet transfer. Non-specific binding was blocked by incubating the membrane in 5% non-fat dry milk in Tris-buffered saline with Tween 20 (25  mmol/L Tris [pH 7.5], 150  mmol/L NaCl, and 0.1% Tween 20) for 1 h, followed by overnight incubation at 4 °C with antibodies against collagen I (Cell Signaling Technology, Beverly, MA, USA), TGF-β_1_ (Abcam), IL-1β (Abcam), ASC (AdipoGen Life Sciences, San Diego, CA, USA), caspase-1 (AdipoGen Life Sciences), NLRP3 (AdipoGen Life Sciences), and actin (Santa Cruz Biotechnology, Santa Cruz, CA, USA). Horseradish peroxidase-conjugated anti-rabbit or anti-mouse IgG (Cell Signaling Technology) was used to detect antibody binding, which was visualized using the iBrightTM CL1500 imaging system (Thermo Fisher Scientific) after treatment with enhanced chemiluminescence reagent (Thermo Fisher Scientific). Densitometric analysis of each band was carried out using the iBright analysis software (Thermo Fisher Scientific). For quantification of specific bands, a square of the same size was drawn around each band for density measurement, and the value was adjusted to the background density near that band. The results are expressed as the relative ratio of the target to the reference protein, with the relative ratio of the target protein of the control group set to 1. 

### 2.9. Statistical Analysis

Statistical analyses were performed using the SigmaPlot v.12 software (Systat, San Jose, CA, USA). Data are expressed as means ± SD. Statistical comparisons were performed by one-way analysis of variance followed by the Tukey or Dunnett test. A value of *p* < 0.05 was considered statistically significant.

## 3. Results

### 3.1. Changes in Body and Organ Weights

Body weight remained constant during the experimental period, and no differences were observed among the groups (Figure 2A). There were no differences in the relative weights of the lung (Figure 2B), spleen (Figure 2C), and thymus (Figure 2D) in DEP-instilled mice compared to the vehicle control mice. However, there were significant differences in the relative weights of these organs in the LPS-instilled group compared to the vehicle control group. In addition, the relative weights of the lung and spleen in the DEP pre-exposed and LPS-instilled groups were higher than those in the LPS-instilled group. The relative weight of the thymus in the DEP pre-exposed and LPS-instilled groups was lower than that in the LPS-instilled group.

### 3.2. DEP Pre-Exposure Exacerbates LPS-Induced Acute Lung 

To investigate the potential role of DEP pre-exposure in LPS-induced ALI, we analyzed the number of inflammatory cells in the BAL fluid and histological changes in the lung tissue following LPS instillation in the DEP pre-exposed group. Total cells and the number of inflammatory cells, including neutrophils and lymphocytes, in the BAL fluid were significantly increased in the DEP and LPS-instilled groups compared to that in the vehicle control group. The number cells in the BAL fluid, especially the total number of cells and that of neutrophils, was significantly increased in the DEP pre-exposed and LPS-instilled groups compared to that in the LPS-induced ALI group (Figure 3A–D). In addition, we examined the H&E and MT stained lung sections of mice in the VC, DEP, LPS, or DEP pre-exposed, and LPS-instilled groups. We observed the accumulation of black particle-laden macrophages in alveoli, and granulomatous inflammation/pulmonary fibrosis in the lung tissue of mice in the DEP-induced group. Infiltration of neutrophils was predominantly observed in the lungs of animals instilled with LPS. The lungs of mice in the DEP pre-exposed and LPS-instilled groups showed markedly enhanced acute inflammation in alveolar/interstitial tissues and granulomatous inflammation/pulmonary fibrosis compared with that in the mice in the DEP or LPS group (Figure 3E). These results were confirmed by histological scoring of infiltration of inflammatory cells as well as of granulomatous inflammation/pulmonary fibrosis (Table 1).

### 3.3. DEP Pre-Exposure Upregulates Pro-Inflammatory Protein Expression in the BAL Fluid of LPS-Instilled Mice

The expression levels of pro-inflammatory molecules, including IL-1β, IL-6, and TNF-α, were measured by ELISA using the BAL fluid, 48 h after LPS instillation. In the DEP group, there were no differences in the levels of these inflammatory proteins, IL-1β, IL-6, and TNF-α compared with the respective levels in the vehicle control. In the LPS group, the levels of all these proteins in the BAL fluid were significantly higher than those in the vehicle control group. In the DEP pre-exposed and LPS-instilled groups, the concentrations of these cytokines in the BAL fluid were significantly increased compared to those in the LPS group (Figure 4A–C).

### 3.4. DEP Pre-Exposure Induces Lung Fibrosis by Upregulating Collagen I and TGF-β_1_ Protein Expression in LPS-Instilled Mice

The histological assessment showed fibrotic changes in the DEP pre-exposed and LPS-instilled group. To confirm these observations, we determined the expression of proteins associated with lung fibrosis, including collagen I and TGF-β_1_, using Western blot analysis with the lung tissue extract, 48 h after LPS instillation. In the DEP and LPS groups, there were no differences in the levels of collagen I and TGF-β_1_ in the lung compared with the respective levels in the vehicle control. In the DEP pre-exposed and LPS-instilled groups, levels of these proteins in the lungs were significantly increased compared with those in the LPS group (Figure 5A,B). These results were confirmed by IHC staining (Figure 5C) and scoring for TGF-β_1_ as a lung fibrosis marker in the lung (see Table 2). TGF-β_1_-positive cells were significantly increased in the lungs of the DEP pre-exposed and LPS-instilled group. Our results indicate that DEP pre-exposure might induce a fibrotic response by upregulating collagen I and TGF-β_1_ expression in LPS-induced ALI.

### 3.5. The Upregulation of IL-17 in the BAL Fluid Contributes to Fibrosis in the Lung of LPS-Instilled Mice Pre-Exposed to DEP

IL-17 plays crucial roles in the pathogenesis of lung diseases, such as COPD, asthma, and fibrosis [16,17] and is involved in the progression of these diseases. Elevated expression of IL-17A also leads to severe progression of injury to the fibro-proliferative phase of the disease [16,17]. There were no differences in the IL-17 levels in the BAL fluid between the DEP and LPS-instilled groups compared with the vehicle control group. Interestingly, the IL-17 concentration in the BAL fluid was significantly increased in the DEP pre-exposed and LPS-instilled group (Figure 6A). These results were confirmed by IHC staining (Figure 6B) and scoring for IL-17 in the lungs (see Table 2). The number of IL-17-positive cells was significantly increased in the lungs of mice in the DEP pre-exposed and LPS-instilled group.

### 3.6. Effect of DEP Pre-Exposure on the NLRP3 Inflammasome in LPS-Induced Acute Lung Injury

The NLRP3 inflammasome is an intracellular sensor that detects a broad range of microbial motifs, endogenous danger signals, and environmental irritants [19,20,21]. The NLRP3 inflammasome, composed of the NLRP3 protein, caspase-1, and apoptosis-associated speck-like protein (ASC), plays a vital role in regulating inflammation [19,20,21]. We determined the expression of NLRP3, caspase-1, ASC, and IL-1β in the lungs of mice in the VC, DEP, LPS, and DEP pre-exposed and LPS-instilled groups. There was no expression of ASC, caspase-1, IL-1β, and NLRP3 upon instillation with DEP compared with that in the VC group. In contrast, instillation with LPS significantly increased the expression of NLRP3 inflammasome components compared with that in the vehicle instillation group. Protein expression was higher in the DEP pre-exposed and LPS-instilled group than in the LPS group (Figure 7).

## 4. Discussion

In this study, to investigate the effect of DEP exposure on LPS-induced ALI and to elucidate the molecular mechanisms underlying the effect, we examined pathological and biological features of ALI in response to DEP pre-exposure by measuring the levels of various proteins, including TNF-α, IL-1β, IL-6, IL-17, NLRP3 inflammasome components, and fibrosis markers in the BAL fluid and lungs. We also analyzed the histological changes in the lung and cellular changes in the BAL fluid using an in vivo experimental system. Our results show that the levels of pro-inflammatory cytokines, NLRP3 inflammasome, and inflammatory cell infiltration were higher in the DEP pre-exposed and LPS-instilled group than in the LPS-induced ALI group. Histological assessments, including H&E and MT staining, showed exacerbated lung inflammation and lung fibrosis in the DEP pre-exposed and LPS-instilled group. Interestingly, we observed that the levels of IL-17, TGF-β_1_, and collagen I were elevated in the lungs of the DEP pre-exposed and LPS-instilled group but not in the DEP or LPS-instilled group.

ALI/ARDS is a significant cause of morbidity and mortality in humans [1,2,3,4,5,6,7,8,9,10]. Infectious etiologies, such as sepsis and pneumonia, are the leading causes of ALI, which is histologically characterized by a severe acute inflammatory response in the lungs and neutrophilic alveolitis [1,10,12,13,14]. The physiological hallmark of ALI/ARDS is the disruption of the alveolar-capillary membrane barrier, leading to the development of noncardiogenic pulmonary edema, in which a proteinaceous exudate floods the alveolar spaces, impairs gas exchange, and precipitates respiratory failure [1,10,11,12,13,14,15]. Accumulating evidence shows that pattern recognition receptors, such as nonendogenous PAMPs and endogenous DAMPs, initiate inflammatory signaling cascades and the release of pro-inflammatory cytokines, such as TNF-α, IL-1β, and IL-8, and induce the production of antibacterial molecules [35,36]. Neutrophils also accumulate in the lungs and release pro-inflammatory cytokines and neutrophil extracellular traps (NETs), which trap pathogens in the extracellular space through NETosis [37,38]. The lung epithelium, specifically the type II alveolar cells, is damaged by these cells and their products, resulting in the disruption of the alveolar–capillary interface and increased pulmonary microvascular permeability. These ALI/ARDS can easily occur after triggering events, such as pneumonia, gastric aspiration, inhalation of smoke and toxic gases, PM, reperfusion, and severe sepsis. Furthermore, epidemiologic studies have shown that PM exposure is associated with increases in respiratory diseases, including ALI/ARDS-related mortality and morbidity [2,5,6,7,8]. Previous studies have shown an association between exposure to air pollution and ALI/ARDS risk. [3,9,10,11,12,13,14]. However, the molecular mechanisms underlying the susceptibility to ALI/ARDS upon PM exposure are not yet completely understood.

In this study, we first selected DEP as one of the major components of PM2.5, and used LPS derived from the cell walls of gram-negative bacteria to generate a mice model of ALI/ARDS, which showed recruitment of inflammatory cells into the lungs with subsequent increases in capillary permeability and neutrophilic alveolitis. We confirmed DEP or LPS-induced pathologic features in the lungs. There were no significant changes in the body weight during the in vivo study. However, increased lung and spleen weights and decreased thymus weight were observed, especially in the LPS-induced group. These results indicate lung damage and alterations in the immune function by DEP or LPS instillation. These damages were more prominent in the LPS group. Corresponding to these results, H&E staining of the lung tissue and cellular changes in the BAL fluid showed slight induction in lung inflammation/granulomatous inflammation and infiltration of DEP-pigmented alveolar macrophages by DEP, and LPS induced moderately acute lung inflammation and neutrophilic alveolitis. We investigated the effect of DEP pre-exposure on ALI with DEP or LPS-induced murine models established in the present study.

We compared the pathological features in the lung of mice in the VC, DEP, LPS, and DEP pre-exposed and LPS-instilled groups. In the DEP pre-exposed and LPS-instilled group, increased lung and spleen weights and decreased thymus weight were markedly observed compared to that in the DEP or LPS-induced group without any significant changes in the body weight during the experimental period. Among these groups, the number of total cells and neutrophils increased the most in the BAL fluid of DEP pre-exposed and LPS-instilled group. The number of these cells was higher in the BAL fluid of DEP pre-exposed and LPS-instilled mice than in LPS-instilled mice. Similar to these results, dominant black pigmented macrophages and slight granulomatous inflammation were observed in the lungs of mice in the DEP group by H&E staining and histological scoring. Acute alveolar/interstitial and neutrophilic infiltration in alveoli was enhanced in the lungs of mice in the DEP pre-exposed and LPS-instilled group compared to that in the LPS group. In particular, we observed significant granulomatous inflammation in the lungs of mice in the DEP pre-exposed and LPS-instilled group. Furthermore, the results of ELISA showed that there was no change in the levels of the pro-inflammatory cytokines, such as IL-1β, IL-6, and TNF-α, in the BAL fluid of mice in the DEP group compared to the respective levels in the VC group. However, LPS-induced increases in pro-inflammatory cytokines in the BAL fluid were higher in the BAL fluid of mice in the DEP pre-exposed and LPS-instilled group. Our results suggest that DEP pre-exposure can exacerbate LPS-induced acute lung inflammation through potent upregulation of pro-inflammatory cytokines and might contribute to fibrotic changes in LPS-induced ALI.

Interestingly, we observed fibrotic changes in only the DEP pre-exposed and LPS-instilled group. Fibrosis is the abnormal formation of excess fibrous connective tissue during chronic inflammation and tissue repair. The excessive deposition of extracellular matrix (ECM) components, especially collagens, is the leading cause of fibrosis. It has been recognized that both innate (macrophages, neutrophils, NK cells, innate lymphoid cells (ILCs), γδT cells, dendritic cells, and NKT cells and mucosal-associated invariant T (MAIT) cells) and adaptive immune cells (T helper (Th) 1, 2, 17, regulatory T cells, follicular helper T cells, and B cells) are important players that perform multiple functions in fibrogenesis [39,40]. In this study, to prove the DEP pre-exposure and LPS-medicated fibrotic phenotype in mice, we examined collagen deposition by MT staining and the levels of TGF-β_1_ and collagen I as central mediators of fibrogenesis in the lungs of mice in the DEP, LPS, and DEP pre-exposed and LPS-instilled groups. Our results show that there was significant collagen deposition in the lungs of mice only in the DEP pre-exposed and LPS-instilled group as evident in the MT staining. The results of Western blot analysis showed increased levels of fibrotic proteins, including collagen I and TGF-β_1_, in the lungs of mice only in the DEP pre-exposed and LPS-instilled group. Consistent with the above results, IHC staining and scoring for TGF-β_1_ showed statistically increased levels of TGF-β_1_ in the lungs of mice only the DEP pre-exposed and LPS-instilled group. Furthermore, our results show that IL-17 cytokine was enhanced in the BAL fluid of mice only in the DEP pre-exposed and LPS-instilled group. IL-17 is a key pro-inflammatory cytokine in Th17 and ILC cells and is actively involved in neutrophilic inflammation and airway remodeling of chronic respiratory conditions (fibrotic response) [41]. The number of IL-17+ cells in endobronchial biopsies of patients with asthma is increased in a manner dependent on the severity of the disease [42]. In addition, bronchiolitis obliterans (OB) patients with lung transplant have increased IL-17 and Th17 differentiating cytokines (IL-1β, IL-6, and IL-23) in the BAL fluid compared with the controls [43]. Moreover, IL-17 levels in the BAL fluid were increased in patients with idiopathic pulmonary fibrosis (IPF) compared to the levels in normal volunteers [43]. Su Mi et al. reported that IL-17A participates in the development and progression of pulmonary fibrosis in both TGF-β_1_–dependent and –independent manner in fibrotic murine models [24]. In other studies, it has been shown that bleomycin produced by γδ T cells induced with IL-17 led to significant neutrophilia and promoted pulmonary fibrosis [23,25]. In addition, IL-17 may promote fibrosis. The potential mechanisms include the ability of IL-17 to increase the synthesis and secretion of collagen from epithelial cells and the promotion of epithelial-mesenchymal transition [24]. These findings suggest that DEP pre-exposure might contribute to fibrotic changes through potent upregulation of IL-17 cytokine-mediated TGF-β_1_ and collagen deposition in LPS-induced ALI.

Furthermore, it has been reported that PM2.5 induces lung inflammation and lung fibrosis by activating the NLRP3 inflammasome [20]. It has also been reported that the NLRP3 inflammasome is involved in H_2_O_2_-induced synthesis of type I collagen, which is mediated by the NF-κB signaling pathway [21]. Additionally, the NLRP3 inflammasome contributes to the development of bleomycin-induced pulmonary fibrosis [44]. The NLRP3 inflammasome is a critical component of innate immunity and contributes to the pathology of human diseases, such as asthma, COPD, and pulmonary inflammation [19,20,21,45,46]. The NLRP3 inflammasome comprises the sensor molecule NLRP3, the adaptor protein ASC, and pro-caspase-1. The priming step is initiated by the ligation of pattern recognition receptors, such as toll-like receptors, by conserved microbial structures, such as LPS, which induces the production of pro–IL-1β (the inactive precursor of IL-1β). This is followed by the second step, which is an activation step in which NLRP3 recruits the adaptor protein ASC and pro-caspase-1 to form the NLRP3 inflammasome assembly (NLRP3–ASC–pro-caspase-1 complex), and the NLRP3 inflammasome activates the caspase-1 cascade and produces the pro-inflammatory cytokine, IL-1β. LPS-primed NLRP3 inflammasome activation has been linked to several inflammatory disorders, including ALI [45,46]. In this study, to investigate whether the NLRP3 inflammasome is activated by DEP or LPS instillation, we examined the expression of the NLRP3 assembly components, including NLRP3, ASC, caspase-1, and IL-1β, in the lungs of mice in the DEP and LPS-instilled groups. Our results showed that there was no change in the levels of the NLRP3 inflammasome-related proteins in the lungs of mice in the DEP group. As expected, these proteins were significantly increased in the lungs of mice in the LPS-instilled group. In the DEP pre-exposed and LPS-instilled group, the expression levels of the NLRP3 inflammasome proteins were higher than in the LPS-instilled group. In addition, the results of ELISA showed that DEP pre-exposure aggravates LPS-induced acute lung inflammation through potent upregulation of pro-inflammatory cytokines. These findings indicate that DEP pre-exposure potently activates LPS-induced NLRP3 inflammasome signaling and might contribute, at least in part, to the enhancement of susceptibility to ALI.

## 5. Conclusions

Our in vivo findings suggest that DEP pre-exposure enhances the susceptibility to LPS-induced ALI. It induces fibrotic changes through the upregulation of the expression of IL-17–derived TGF-β_1_ and collagen I, at least in part, and enhances LPS-induced NLRP3 inflammasome activation in mice.

## Figures and Tables

**Figure 1 biomolecules-11-00067-f001:**
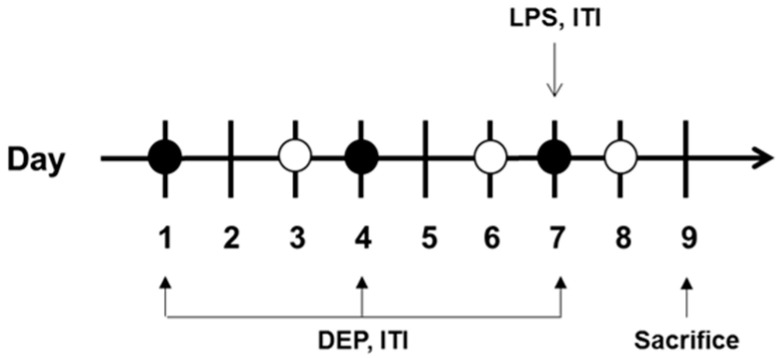
Diagram showing the in vivo experimental protocol. DEP, diesel exhaust particulate; LPS, lipopolysacharide; ITI, intratracheal injection.

**Figure 2 biomolecules-11-00067-f002:**
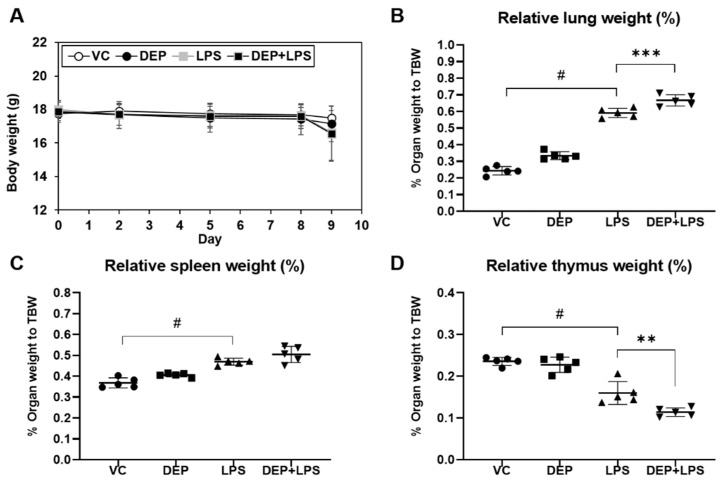
Changes in (**A**) body weight and (**B**–**D**) relative organ weight in vehicle control (VC), DEP, LPS, DEP pre-exposed, and LPS-instilled (DEP+LPS) groups. Relative weights of the (**B**) lung, (**C**) spleen, and (**D**) thymus were calculated using the following formula: relative organ weight = organ weight (g)/final body weight (g) × 100%. Data represent means ± SD (n = 5 per group). ^#^
*p* < 0.05 vs. VC, ** *p* < 0.01 or *** *p* < 0.001 vs. LPS group.

**Figure 3 biomolecules-11-00067-f003:**
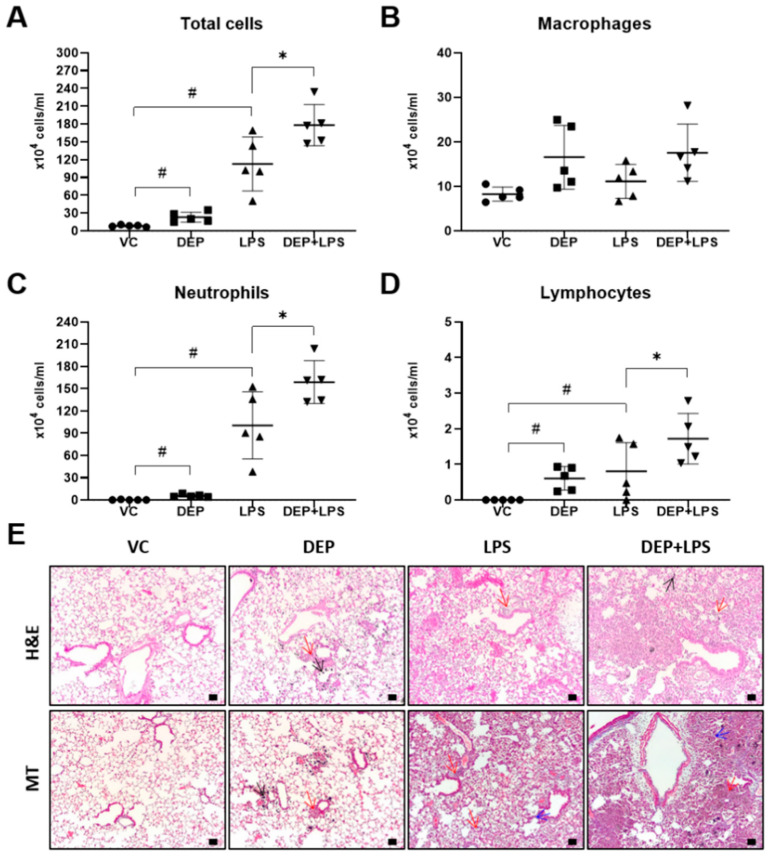
(**A**–**D**) Cellular changes in the bronchoalveolar lavage (BAL) fluid and (**E**) representative hematoxylin and eosin (H&E) and Masson’s trichrome-stained images of the lung sections from the vehicle control (VC), DEP, LPS, and DEP pre-exposed and LPS-instilled (DEP + LPS) groups. Black, red, and blue arrows indicate particle-pigmented alveolar macrophages, inflammatory infiltration, and collagen deposition, respectively. Scale bars = 50 μm. Data represent means ± SD (n = 5 per group). ^#^
*p* < 0.05 vs. VC, * *p* < 0.05 vs. LPS group.

**Figure 4 biomolecules-11-00067-f004:**
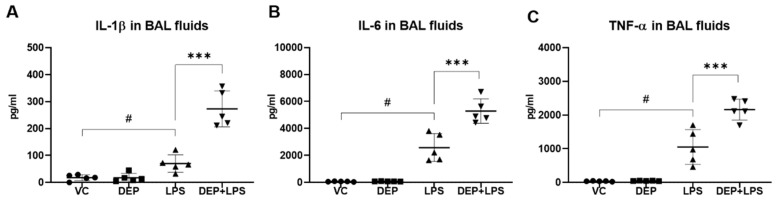
Protein levels of (**A**) IL-1β, (**B**) IL-6, and (**C**) TNF-α in the bronchoalveolar (BAL) fluid from mice in the vehicle control (VC), DEP, LPS, and DEP pre-exposed and LPS-instilled (DEP+LPS) groups. Data represent means ± SD (n = 5 per group). ^#^
*p* < 0.05 vs. VC, *** *p* < 0.001 vs. LPS group.

**Figure 5 biomolecules-11-00067-f005:**
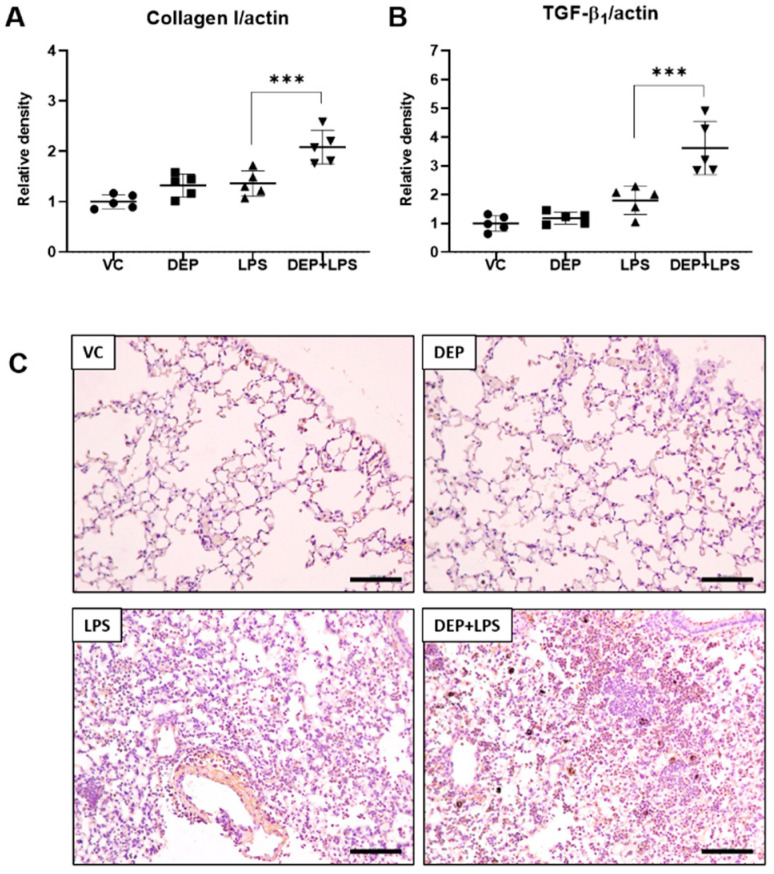
Protein levels of (**A**) collagen I and (**B**) TGF-β_1_ and representative images of immunohistochemistry for TGF-β_1_ ((**C**), brown color) in the lung tissue sections from mice in the VC, DEP, LPS, and DEP pre-exposed and LPS-instilled (DEP+LPS) groups. Scale bars = 500 nm. Data represent means ± SD (n = 5 per group). *** *p* < 0.001 vs. LPS group.

**Figure 6 biomolecules-11-00067-f006:**
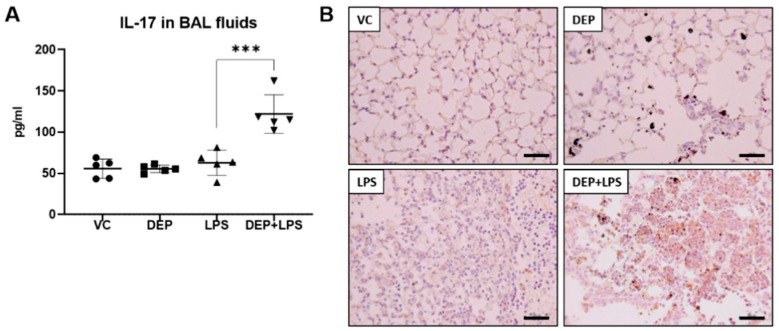
Protein levels of (**A**) IL-17 in the bronchoalveolar (BAL) fluid and (**B**) representative images of immunohistochemistry (IHC) for IL-17 (brown color) in the lung tissue sections from mice in the VC, DEP, LPS, and DEP pre-exposed and LPS-instilled (DEP+LPS) groups. Scale bars = 200 nm. Data represent means ± SD (n = 5 per group) *** *p* < 0.001 vs. LPS group.

**Figure 7 biomolecules-11-00067-f007:**
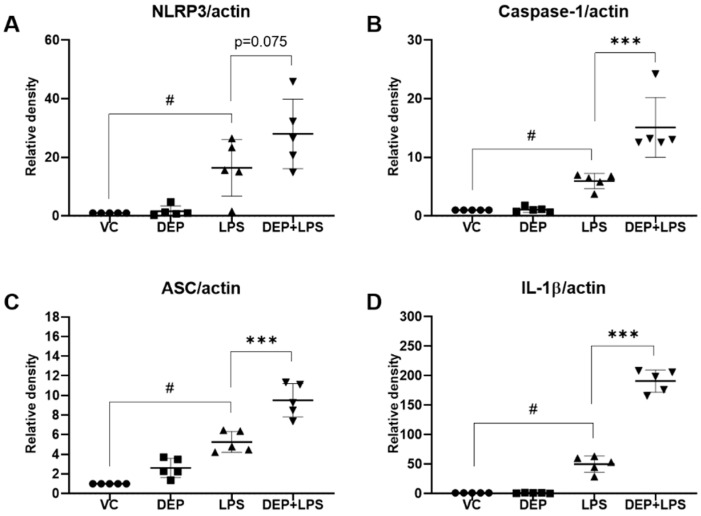
Protein levels of (**A**) NLRP3, (**B**) caspase-1, (**C**) ASC, and (**D**) IL-1β in the lung tissue of mice in the vehicle control (VC), DEP, LPS, and DEP pre-exposed and LPS-instilled (DEP+LPS) groups. Data represent means ± SD (n = 5 per group). ^#^
*p* < 0.05 vs. VC, *** *p* < 0.001 vs. LPS group.

**Table 1 biomolecules-11-00067-t001:** Histological scores for the lung of mice in different groups.

Group	VC	DEP	LPS	DEP + LPS
Black pigmented macrophages	0	3.2 ± 0.45 ^##^	0	3.4 ± 0.55 **
Granulomatous inflammation/pulmonary fibrosis	0	0.4 ± 0.55	0.4 ± 0.55	1.8 ± 0.45 **
Acute inflammation, alveolar/interstitial	0	0	2.6 ± 1.52 ^##^	3.8 ± 0.45
Infiltrate, neutrophilic cells, alveolar	0	0	3.2 ± 0.84 ^##^	3.6 ± 0.55

1: minimal; 2: slight; 3: moderate; 4: severe. Data are presented as means ± SD for five mice per group. DEP, diesel exhaust particulate. ^##^
*p* < 0.001 vs. VC; ** *p* < 0.001 vs. LPS.

**Table 2 biomolecules-11-00067-t002:** Immunohistochemical scores for TGF-β_1_ and IL-17A.

Group	VC	DEP	LPS	DEP + LPS
IL-17A	0	0	1.2 ± 0.71	2.2 ± 0.45 **
TGF-β_1_	0	0.2 ± 0.45	0.2 ± 0.45	3.4 ± 0.55 **

1: minimal; 2: slight; 3: moderate; 4: severe. Data are presented as means ± SD for five mice per group. DEP, diesel exhaust particulate. ** *p* < 0.001 vs. LPS.

## Data Availability

The datasets used and/or analyzed during the current study are available from the corresponding author on reasonable request.

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
