# Peer review of "Diesel Exhaust Particulates Enhances Susceptibility of LPS-Induced Acute Lung Injury through Upregulation of the IL-17 Cytokine-Derived TGF-β1/Collagen I Expression and Activation of NLRP3 Inflammasome Signaling in Mice"

_biomolecules, 2021, doi:10.3390/biom11010067_

Round 1

Reviewer 1 Report

In this manuscript by Kim et al., the authors investigate the biological mechanism of interaction between particular matter air pollution exposure and acute pulmonary inflammation induced by LPS. By use of an in vivo LPS induced acute lung injury model, the authors investigated the effect of pre-exposure to diesel exhaust particulate matter on response to LPS by intratracheal injection. They observed that pre-treatment with DEP synergistically enhances the acute lung injury in response to LPS, and they highlight an interesting observation that pretreatment with DEP appears to particularly modify NLRP3 inflammasome components, TGFbeta, and IL17 with gross pathological changes including increased pulmonary fibrosis. This is an interesting study with a novel finding that appears well conducted and has merit for publication. There are several factors that should be addressed prior to publication.

Major:

  • In reading this manuscript, to me it does not appear that the overall rationale for the study is clearly laid out. The authors introduce the serious clinical condition of acute lung injury and ARDS and then introduce the public health concern of particular matter air pollution and its association on a population level for emergency department visits and hospitalizations for respiratory infections. These are very different things, ARDS being a somewhat rare clinical condition that carries extremely high risk of mortality and ED visits for respiratory infections being common and carry much lower mortality risk. It is important to note that ARDS is not what has been reported to be associated with PM2.5 in the cited studies evaluating ED visits and hospitalizations. Importantly, of the large observational studies on mortality and PM2.5 air pollution, the mortality risk appears to be predominantly secondary to cardiovascular death. It will be important for the authors lay out and clearly state that they hypothesize that air pollution exposure would worsen ARDS and if there is clinical data to support an association with ARDS then this should be presented (the authors citation of Strosnider et al. does not look at ARDS).

  • The authors do not appear to cite previous work investigating the interaction between PM2.5 exposure and LPS induced pulmonary inflammation. Although the authors are bringing an interesting observation to this biological question with an emphasis on IL17 and the inflammasome, this is not a particularly new question and there is a sizeable amount of literature already out there on this question. It would be helpful to the reader to be able to understand what has been done previously on this topic and put this manuscript into context with what has already been done. For example, there is a very recent paper by Costa et al. in Scientific Reports examining a very similar question.

https://pubmed.ncbi.nlm.nih.gov/32943719/

Other references to consider citing:

Inoue, H. et al. Ultrastructural changes of the air-blood barrier in mice after intratracheal instillation of lipopolysaccharide and ultrafine carbon black particles. Exp. Toxicol. Pathol. 61, 51–58. https://doi.org/10.1016/j.etp.2007.10.001 (2009).

Roberts, E. S., Richards, J. H., Jaskot, R. & Dreher, K. L. Oxidative stress mediates air pollution particle-induced acute lung injury and molecular pathology. Inhal. Toxicol. 15, 1327–1346. https://doi.org/10.1080/08958370390241795 (2003).

Takano, H. et al. Diesel exhaust particles enhance lung injury related to bacterial endotoxin through expression of proinflammatory cytokines, chemokines, and intercellular adhesion molecule-1. Am. J. Respir. Crit. Care Med. 165, 1329–1335. https://doi.org/10.1164/rccm.2108122 (2002).

Yanagisawa, R. et al. Enhancement of acute lung injury related to bacterial endotoxin by components of diesel exhaust particles. Thorax 58, 605–612. https://doi.org/10.1136/thorax.58.7.605 (2003).

Yanagisawa, R. et al. Complementary DNA microarray analysis in acute lung injury induced by lipopolysaccharide and diesel exhaust particles. Exp. Biol. Med. 229, 1081–1087. https://doi.org/10.1177/153537020422901013 (2004)

  • Page 3 line 97. Reports that mice were divided into 5 weight-matched experimental groups, yet only 4 groups are presented in the figures. Are there 2 different vehicle control groups each with 5 mice (one with DW and one with NS)? This discrepancy needs to be addressed more clearly. One way to help present this would be for Figure 1 to add each treatment arm to the figure to better demonstrate treatment protocol.

Minor:

  • Page 3 line 101-104, the microgram symbol does not appear in the PDF, likely formatting issue but make sure the correct units are displayed

  • Page 4 line 150 – typo. “abnti’ should be anti

  • Figure 3, would recommend redemonstrating the number of cells for macrophages, eosinophils, and lymphocytes with a Y axis that demonstrates the changes across groups. Particularly if you are demonstrating a treatment related effect on lymphocytic infiltration.

  • Page 11 line 376 – ‘collagen was significantly deposed’ should be ‘there was significant collagen deposition’ Deposed is not the past participle of deposit.

  • There is a trend where journals are moving towards requiring bar graphs to be presented with each data point depicted as a dot plot. There is an advantage to this as it clearly shows the reader what the variability in the data is, rather than masking this with a mean and SE. It appears these figures are done with Graphpad prism software, it is easy to change the presentation to show each individual data point.

Author Response

Manuscript ID: biomolecules-1051009

Responses to the Reviewers' Comments

We appreciate very much the editor and the reviewers for the constructive comments. We also thank the editor and the reviewers for the effort and time put into the review of the manuscript. Each comment has been carefully considered point by point and responded. Responses to the reviewers and changes in the revised manuscript are as follows. Thank you for your consideration. I am looking forward to your positive response.

# Reviewer 1: Thank you for your thoughtful and thorough review of our manuscript.

Comments and Suggestions for Authors:

In this manuscript by Kim et al., the authors investigate the biological mechanism of interaction between particular matter air pollution exposure and acute pulmonary inflammation induced by LPS. By use of an in vivo LPS induced acute lung injury model, the authors investigated the effect of pre-exposure to diesel exhaust particulate matter on response to LPS by intratracheal injection. They observed that pre-treatment with DEP synergistically enhances the acute lung injury in response to LPS, and they highlight an interesting observation that pretreatment with DEP appears to particularly modify NLRP3 inflammasome components, TGFbeta, and IL17 with gross pathological changes including increased pulmonary fibrosis. This is an interesting study with a novel finding that appears well conducted and has merit for publication. There are several factors that should be addressed prior to publication.

Major points:

  1. In reading this manuscript, to me it does not appear that the overall rationale for the study is clearly laid out. The authors introduce the serious clinical condition of acute lung injury and ARDS and then introduce the public health concern of particular matter air pollution and its association on a population level for emergency department visits and hospitalizations for respiratory infections. These are very different things, ARDS being a somewhat rare clinical condition that carries extremely high risk of mortality and ED visits for respiratory infections being common and carry much lower mortality risk. It is important to note that ARDS is not what has been reported to be associated with PM2.5 in the cited studies evaluating ED visits and hospitalizations. Importantly, of the large observational studies on mortality and PM2.5 air pollution, the mortality risk appears to be predominantly secondary to cardiovascular death. It will be important for the authors lay out and clearly state that they hypothesize that air pollution exposure would worsen ARDS and if there is clinical data to support an association with ARDS then this should be presented (the authors citation of Strosnider et al. does not look at ARDS).

Response: Thank you for reviewer’s comment. To clarify the hypothesis that exposure to air pollution can exacerbate ARDS, we provided references on the association between air pollutant and ARDS. Also, we revised the introduction part (page 2, line 50-57).

Reference

  1. Moazed, F.; Calfee, C. S. Environmental Risk Factors for ARDS, Clin Chest Med. 2014, 35, 625–637. doi:10.1016/j.ccm.2014.08.003.
  2. Lin, H.; Tao, J.; Kan, H.; Qian, Z.; Chen, A.; Du, Y.; Liu, T.; Zhang, Y.; Qi, Y.; Ye, J.; Li, S.; Li, W.; Xiao, J.; Zeng, W.; Li, X.; Stamatakis, K. A.; Chen, X.; Ma, W. Ambient particulate matter air pollution associated with acute respiratory distress syndrome in Guangzhou, China. J Expo Sci Environ Epidemiol. 2018, 28, 392-399. doi: 10.1038/s41370-018-0034-0.
  3. Rhee, J.; Dominici, F.; Zanobetti, A.; Schwartz, J.; Wang, Y.; Di, Q.; Balmes, J.; Christiani, D. C. Impact of Long-Term Exposures to Ambient PM 2.5 and Ozone on ARDS Risk for Older Adults in the United States. Chest. 2019, 156, 71-79. doi:10.1016/j.chest.2019.03.017.
  4. Reilly, J. P.; Zhao, Z.; Shashaty, M. G. S.; Koyama, T.; Christie, J. D.; Lanken, P. N.; Wang, C.; Balmes, J. R.; Matthay, M. A.; Calfee, C. S.; Ware, L. B. Low to Moderate Air Pollutant Exposure and Acute Respiratory Distress Syndrome after Severe Trauma. Am J Respir Crit Care Med. 2019, 199, 62–70. doi:10.1164/rccm.201803-0435OC
  5. Rush, B.; McDermid, R. C.; Celi, L. A.; Walley, K. R.; Russell, J. A.; Boyd, J. H. Association between chronic exposure to air pollution and mortality in the acute respiratory distress syndrome. Environ Pollut. 2017, 224, 352-356. doi:10.1016/j.envpol.2017.02.014

  1. The authors do not appear to cite previous work investigating the interaction between PM2.5 exposure and LPS induced pulmonary inflammation. Although the authors are bringing an interesting observation to this biological question with an emphasis on IL17 and the inflammasome, this is not a particularly new question and there is a sizeable amount of literature already out there on this question. It would be helpful to the reader to be able to understand what has been done previously on this topic and put this manuscript into context with what has already been done. For example, there is a very recent paper by Costa et al. in Scientific Reports examining a very similar question.

https://pubmed.ncbi.nlm.nih.gov/32943719/

Other references to consider citing:

Inoue, H. et al. Ultrastructural changes of the air-blood barrier in mice after intratracheal instillation of lipopolysaccharide and ultrafine carbon black particles. Exp. Toxicol. Pathol. 61, 51–58. https://doi.org/10.1016/j.etp.2007.10.001 (2009).

Roberts, E. S., Richards, J. H., Jaskot, R. & Dreher, K. L. Oxidative stress mediates air pollution particle-induced acute lung injury and molecular pathology. Inhal. Toxicol. 15, 1327–1346. https://doi.org/10.1080/08958370390241795 (2003).

Takano, H. et al. Diesel exhaust particles enhance lung injury related to bacterial endotoxin through expression of proinflammatory cytokines, chemokines, and intercellular adhesion molecule-1. Am. J. Respir. Crit. Care Med. 165, 1329–1335. https://doi.org/10.1164/rccm.2108122 (2002).

Yanagisawa, R. et al. Enhancement of acute lung injury related to bacterial endotoxin by components of diesel exhaust particles. Thorax 58, 605–612. https://doi.org/10.1136/thorax.58.7.605 (2003).

Yanagisawa, R. et al. Complementary DNA microarray analysis in acute lung injury induced by lipopolysaccharide and diesel exhaust particles. Exp. Biol. Med. 229, 1081–1087. https://doi.org/10.1177/153537020422901013 (2004)

Response: Thank you for reviewer’s comment. We added previous studies about the interaction between PM2.5 exposure and LPS induced pulmonary inflammation and revised the introduction part (page 2, 58-66).

Reference

  1. Yanagisawa, R.; Takano, H.; Inoue K.;, Ichinose T.;, Sadakane, K.; Yoshino, S.; Yamaki, K.; Kumagai, Y.; Uchiyama, K.; Yoshikawa, T.; Morita M. Enhancement of acute lung injury related to bacterial endotoxin by components of diesel exhaust particles. Thorax. 2003, 58, 605–612. doi: 10.1136/thorax.58.7.605 (2003).
  2. Takano, H.; Yanagisawa, R.; Ichinose, T.; Sadakane, K.; Yoshino, S.; Yoshikawa, T.; Morita M. Diesel exhaust particles enhance lung injury related to bacterial endotoxin through expression of proinflammatory cytokines, chemokines, and intercellular adhesion molecule-1. Am. J. Respir. Crit. Care Med. 2002, 165, 1329–1335. doi: 10.1164/rccm.2108122 (2002).
  3. Yanagisawa, R.; Takano, H.; Inoue, K.; Ichinose, T.; Yoshida, S.; Sadakane, K.; Takeda, K.; Yoshino, S.; Yamaki, K.; Kumagai, Y.; Yoshikawa T. Complementary DNA microarray analysis in acute lung injury induced by lipopolysaccharide and diesel exhaust particles. Exp. Biol. Med. 2004, 229, 1081–1087. doi: 10.1177/153537020422901013 (2004)
  4.  
  1. Page 3 line 97. Reports that mice were divided into 5 weight-matched experimental groups, yet only 4 groups are presented in the figures. Are there 2 different vehicle control groups each with 5 mice (one with DW and one with NS)? This discrepancy needs to be addressed more clearly. One way to help present this would be for Figure 1 to add each treatment arm to the figure to better demonstrate treatment protocol. Mice in the vehicle control (VC) group received the same volume of distilled water (DW) and normal saline and served as the DEP and LPS control, respectively, during experimental periods.

Response: In our study, mice were divided into 4 weight-matched experimental groups. So, we revised (page 3, line 108). Also, we revised study protocol part to clarify study design. We used distilled water (DW) and normal saline as the DEP and LPS vehicle control, respectively. Mice in the vehicle control (VC) group received 50 μL distilled water (DW) as the DEP control and instilled 50 μL normal saline as LPS control. The DEP group received 100 μg DEP (SRM 2975; National Institute of Standards and Technology, Gaithersburg, MD, USA) dispersed in 50 μL DW and instilled 50 μL normal saline. The LPS group received 50 μL DW and instilled 20 μg LPS (Sigma-Aldrich, St. Louis, MO, USA) dissolved in 50 μL of normal saline to induce ALI. DEP pre-exposure and LPS-instilled groups were pretreated with 100 μg DEP in 20 μg LPS-induced mice (page 3, line 110-116).

Minor points:

  1. Page 3 line 101-104, the microgram symbol does not appear in the PDF, likely formatting issue but make sure the correct units are displayed

Response: We revised them (page 3, line 110-116)

  1. Page 4 line 150 – typo. “abnti’ should be anti

Response: We deleted “b” (page 4, line 161).

  1. Figure 3, would recommend redemonstrating the number of cells for macrophages, eosinophils, and lymphocytes with a Y axis that demonstrates the changes across groups. Particularly if you are demonstrating a treatment related effect on lymphocytic infiltration.

Response: We revised according to review’s comment (page 6, figure 3).

  1. Page 11 line 376 – ‘collagen was significantly deposed’ should be ‘there was significant collagen deposition’ Deposed is not the past participle of deposit.

Response: We revised according to reviewer’s comment (page 11, ling 387-388).

  1. There is a trend where journals are moving towards requiring bar graphs to be presented with each data point depicted as a dot plot. There is an advantage to this as it clearly shows the reader what the variability in the data is, rather than masking this with a mean and SE. It appears these figures are done with Graphpad prism software, it is easy to change the presentation to show each individual data point

Response: We changed bar graphs to dot plots in figures.

Reviewer 2 Report

The manuscript describes the effects of pre-exposure to diesel exhaust particulates on LPS-induced acute lung injury in mice.

The work is well conducted and well exposed.

In my opinion the manuscript could be published in this journal.

  • line 150 correct abnti IL17A
  • line 198/199 correct inflammation in mice
  • line 228 please move the table 2 after line 255 (citation of the table in the text) to improve the readability of the text
  • please explain the choice of DEP and LPS doses and time of administration

Author Response

Manuscript ID: biomolecules-1051009

Responses to the Reviewers' Comments

We appreciate very much the editor and the reviewers for the constructive comments. We also thank the editor and the reviewers for the effort and time put into the review of the manuscript. Each comment has been carefully considered point by point and responded. Responses to the reviewers and changes in the revised manuscript are as follows. Thank you for your consideration. I am looking forward to your positive response.

# Reviewer 2: Thank you for your thoughtful and thorough review of our manuscript.

Comments and Suggestions for Authors:

The manuscript describes the effects of pre-exposure to diesel exhaust particulates on LPS-induced acute lung injury in mice. The work is well conducted and well exposed. In my opinion the manuscript could be published in this journal.

Minor points:

  1. line 150 correct abnti IL17A

Response: We deleted “b” (page 4, line 161).

  1. line 198/199 correct inflammation in mice

Response: We deleted “inflammation in mice” (page 5, line 209).

  1. line 228 please move the table 2 after line 255 (citation of the table in the text) to improve the readability of the text

Response: We moved the table 2 to line 267 (page 8, table 2).

  1. please explain the choice of DEP and LPS doses and time of administration

Response: We referenced various papers of DEP-instilled animal model (1-3) and selected DEP dose (100 μg). In DEP-induced mice, neutrophilic inflammation were observed in bronchoalveolar lavage fluid by Diff Quik staining and histological changes were minimally observed in lung tissues by H&E staining (4, 5). We thought that these inflammatory level by DEP might synergistically enhance acute lung injury. In case of LPS dose, lethal dose of LPS by intratracheal injection is 25 mg/kg in mice (6). In fact, LPS dose are selected in wide dose ranges (below lethal dose) to induce acute lung injury in mice. Especially, LPS 10-100 μg LPS dose were used in many studies. We referenced various papers about ALI-induced animal models. We selected LPS dose (20 μg) that protective effects of drugs were well observed in mice with ALI-characterized pathological changes (7-9).

Reference

  1. Sharen Provoost, Tania Maes, Monique A. M. Willart, Guy F. Joos, Bart N. Lambrecht and Kurt G. Tournoy., Diesel Exhaust Particles Stimulate Adaptive Immunity by Acting on Pulmonary Dendritic Cells. J Immunol. 2009, doi:10.4049/jimmunol.0902564
  2. Smitha Kumar, Guy Joos, Louis Boon, Kurt Tournoy, Sharen Provoost1 & Tania Maes. Role of tumor necrosis factor–α and its receptors in diesel exhaust particle-induced pulmonary inflammation. Scientific reports. 7, 11508. doi:10.1038/s41598-017-11991-7.
  3. Sharen Provoost, Tania Maes, Nele S. Pauwels, Tom Vanden Berghe, Peter Vandenabeele, Bart N. Lambrecht, Guy F. Joos and Kurt G. Tournoy. NLRP3/Caspase-1-Independent IL-1b Production Mediates Diesel Exhaust Particle-Induced Pulmonary Inflammation. J Immunol. 2011, 187:3331-3337. doi: 10.4049/jimmunol.1004062
  4. Kim, D. I.; Song, M.K.; Kim, S.H.; Park, C. Y.; Lee, K. TF-343 alleviates diesel exhaust particulate-induced lung inflammation via modulation of nuclear factor-κB signaling. J. Immunol. Res. 2019, Article ID 8315845. doi: 10.1155/2019/8315845
  5. Kim, D. I.; Song, M. K.;, Kim, H.I.; Han, K. M.; Lee, K. Diesel Exhaust Particulates Induce Neutrophilic Lung Inflammation by Modulating Endoplasmic Reticulum Stress-Mediated CXCL1/KC Expression in Alveolar Macrophages. Molecules. 2020, 25, 6046; doi:10.3390/molecules25246046
  6. Hui-Hui Yang, et al. A COX-2/sEH dual inhibitor PTUPB alleviates lipopolysaccharide-induced acute lung injury in mice by inhibiting NLRP3 inflammasome activation. Theranostics. 2020, 10, 11, 4749-4761
  7. Yamada, M.; Kubo, H.; Kobayashi, S.; Ishizawa, K.; Numasaki, M.; Ueda, S.; Suzuki, T.; Sasaki, H. Bone Marrow-Derived Progenitor Cells Are Important for Lung Repair after Lipopolysaccharide-Induced Lung Injury. J Immunol. 2004, 172:1266-1272. doi: 10.4049/jimmunol.172.2.1266
  8. Bozinovski, S.; Jones, J.; Beavitt, S.; Cook, A. D.; Hamilton, J. A.; Anderson, G. P. Innate immune responses to LPS in mouse lung are suppressed and reversed by neutralization of GM-CSF via repression of TLR-4. Am J Physiol Lung Cell Mol Physiol. 2004, 286, L877–L885. doi: 10.1152/ajplung.00275.2003
  9. Zhao, M.; Du, J. Anti-inflammatory and protective effects of D-carvone on lipopolysaccharide (LPS)-induced acute lung injury in mice. Journal of King Saud University–Science. 2020, 32, 1592–1596. ttps://doi.org/10.1016/j.jksus.2019.12.016